# Integrating economic measures of adaptation effectiveness into climate change interventions: A case study of irrigation development in Mwea, Kenya

**Daiju Narita**[1,2,3]*, **Ichiro Sato**[4], **Daikichi Ogawada**[5], **Akiko Matsumura**[5]

**1** Graduate School of Arts and Sciences, University of Tokyo, Tokyo, Japan, **2** JICA Ogata Research Institute, Tokyo, Japan, **3** Kiel Institute for the World Economy, Kiel, Germany, **4** Japan International Cooperation Agency (JICA), Sendai, Japan, **5** Nippon Koei Co., Ltd, Tsukuba, Japan

* daiju.narita@global.c.u-tokyo.ac.jp

## Abstract

As climate change adaptation is becoming a recognized policy issue, the need is growing for quantitative economic evaluation of adaptation-related public investment, particularly in the context of climate finance. Funds are meant to be allocated not to any types of beneficial investments with or without climate change but to projects regarded as effective for climate change adaptation based on some metrics. But attempts at such project-specific evaluation of adaptation effects are few, in part because such assessments require an integration of various types of simulation analyses. Against this background, we conduct a case study of a Kenyan irrigation development project using a combination of downscaled climate data, run-off simulations, yield forecasting, and local socioeconomic projections to examine the effects of interventions specifically attributable to climate change adaptation, i.e., how much irrigation development can reduce the negative effects of climate change in the future. The results show that despite the uncertainties in precipitation trends, increased temperatures due to climate change have a general tendency to reduce rice yields, and that irrigation development will mitigate income impacts from the yield loss–for example, for the median scenario, the household income loss of 6% in 2050 due to climate change without irrigation development is flipped to become positive with the project. This means that the irrigation development project will likely be effective as a means for climate change adaptation.

**Data Availability Statement:** Base data for the analysis are found in JICA and Nippon Koei (2018). Economic Evaluation of Adaptation Measures to

## 1. Introduction

As the effects of global climate change have become more evident in various parts of the world, the need for adaptation to these changes has become more apparent. Climate change adaptation necessitates public planning and investment concerning the provision of public goods [1], and as such, there are growing demands for international assistance for adaptation actions in developing countries that are particularly vulnerable to the impact of climate change. UNEP-DTU [2] estimates that annual costs in 36 developing countries to meet their

Climate Change under Uncertainty Phase 2. Tokyo: JICA (in Japanese) – which is mentioned in the reference list of the manuscript. The document is available upon request, by contacting the JICA Global Environment Department through the JICA library (https://libportal.jica.go.jp/library/public/Index.html). Data necessary for figure reproductions are deposited on the following figshare repository site. https://doi.org/10.6084/m9.figshare.13147598.v3.

**Funding:** This study is funded by JICA and the JICA Ogata Research Institute (JICA-RI), which also provided various logistical support. Daiju Narita acknowledges financial support from JSPS Kakenhi (Grant Numbers: 17KT0066 and 17K00677) for his research activities. Besides logistical support, the funders had no role in study design, data collection and analysis, decision to publish, or preparation of the manuscript.

**Competing interests:** The authors have read the journal's policy and have the following competing interests: DO and AM are paid employees of commercial engineering company, Nippon Koei Co. (https://www.n-koei.co.jp/english/). There are no patents, products in development or marketed products associated with this research to declare. This does not alter our adherence to PLOS ONE policies on sharing data and materials.

adaptation targets will range from US$140 to 300 billion by 2030, which is two-to-three times higher than the previous estimates of future adaptation costs and will further increase to US$280–500 billion by 2050. Responding to such growing demands, international funds and institutions of development aid and finance intend to strengthen their support for adaptation action in those countries. These flows of funds constitute climate finance, i.e., the financial resources that are paid to support climate change mitigation and adaptation activities. Just like any other types of project finance, climate finance needs to be allocated to potentially successful projects based on some performance metrics.

Against this background, the international climate and development community is starting to recognize the necessity for a methodology and metrics to quantitatively and financially assess climate change adaptation measures and to compare the adaptation effectiveness of different project options for desirable allocation of funding resources and for project planning. Such a methodology is required also to review the progress of and draw lessons from adaptation actions, and furthermore, it would ensure accountability for the resources allocated to them.

However, conventional methods of project evaluation are inadequate for such assessment because it needs to integrate various sets of data outside of the conventional realm of financial analysis, such as those of climate modeling and hydrological simulations. Another difficulty for such evaluation lies in the fact that the impacts of climate change are highly uncertain due to the long time spans of these changes and a continuing lack of knowledge on the mechanisms of climate and other natural systems [3,4]. Indeed, an expert group of the United Nations Framework Convention on Climate Change (UNFCCC) indicates that the "nature of adaptation, including its long timescales, the uncertainty associated with its impacts and its context-specificity, and difficulties in setting baselines and targets and the consequent lack of common metrics to measure the reduction of vulnerability or the enhancement of adaptive capacity all constrain reviewing the adequacy and effectiveness of adaptation" (Adaptation Committee and Least Developed Countries Expert Group, FCCC/SB/2017/2/Add.1–FCCC/SBI/2017/14/Add.1, dated in September 2017). So far, no established methodology of project-based financial evaluation of adaptation projects exists given the challenges described above.

Our study aims to respond to such methodological needs in the practice of climate change finance in terms of the economic evaluation of climate change adaptation-related projects subject to future uncertainty. To this end, we demonstrate a quantitative evaluation of an irrigation development project in Kenya in terms of adaptation effectiveness. Specifically, we examine and quantify the benefits of irrigation development directly attributable to climate change adaptation, i.e., how much irrigation development can reduce the negative effects of climate change on local farming in the future. We also estimate the *total* benefits of irrigation under climate change, on which the existing studies generally focus, and contrast the two types of values. Our assessment is cross-disciplinary in synthesizing modeling frameworks from climatology, hydrology, agronomy and decision science, and we offer clear descriptions of the climatic, hydrological, agricultural, and economic models we use. The case was chosen as an example of climate change adaptation in general but has significance of its own. Africa is known to be one of the most vulnerable regions in terms of agriculture from climate change [5]. At the same time, a large potential for irrigation exists in Africa and also in Kenya specifically [6,7], and promotion of irrigation could mitigate the negative effects of climate change in the region [8].

Irrigation, which enhances and stabilizes water supply for farming, or makes farming "climate-proof," is often argued as a major means of climate change adaptation [9]. Various global or regional-level assessments have already been made on climate change impacts and adaptation regarding African agriculture using process-based and statistical models [5,10–14], along

with some country-level studies with a similar scope in other regions [15,16]. Also, a number of survey-based economic studies have examined actual adaptation practices and perceptions in the face of the climate risks for African farmers (reviewed by [17]). However, these studies only assess either the general agricultural impacts of climate change or the total agricultural benefits of irrigation under climate change, in other words, they do not isolate and quantify the precise adaptation effects of irrigation interventions for reducing climate change-related yield and income losses, as formulated by [18]. To the authors' knowledge no economic studies of the assessment of climate change adaptation exist, in Africa or elsewhere, in the form of a project-specific evaluation of irrigation development identifying the isolated effects of irrigation on climate change adaptation. Apart from practical usefulness, locality-specific analyses should also offer valuable insights for the general understanding of climate change adaptation as they can reflect realistic institutional arrangements and the particular socioeconomic and environmental conditions of those localities [19].

In principle, the effectiveness of an infrastructure project as a means of climate change adaptation should be evaluated as the avoided loss caused by climate change, expressed in a monetary unit if it is to be integrated into the economic analysis of the project. A major difficulty in such an evaluation is the representation of uncertainty associated with climate change. In our study, we incorporate the uncertainties of future climatic and economic conditions by considering a large number of scenarios without any differentiation of their likelihood (i.e., no use of probabilistic weights), an approach consistent with the Robust Decision Making (RDM) method, a methodological framework of decision-making support under deep uncertainty [20]. In brief, RDM is an approach to analyze decision making for uncertain problems and deals with running simulation models numerous times to stress-test proposed decisions against a wide range of plausible futures.

This paper discusses the simulation methods and the basic results of our integrative modeling of irrigation impacts; a systematic evaluation of the impact of uncertainty on our simulation results based on the RDM framework will be presented in a different paper (Narita et al. in preparation). The results of our simulation analysis can directly be used as a cost-benefit assessment and thus is relatable to the concerns of the climate finance community mentioned above.

## 2. Methods

### 2.1. The study area and the Mwea Irrigation Development Project

The Mwea Irrigation Development Project is an irrigation infrastructure development project in central Kenya undertaken by Kenya's National Irrigation Board (NIB) with a loan provided by the Japan International Cooperation Agency (JICA). Fig 1 shows a map of the Mwea area. The project consists of the construction of an irrigation dam (on the Thiba River, see Fig 1), the construction of a new irrigation canal (Link Canal III), and the improvement of existing canals and irrigation areas. Mwea is located approximately 100km northeast of Nairobi (Latitude 0°41' S and Longitude 37°20' E) at an elevation of 1160m above sea level. The local climate is tropical with two rainy seasons, a long rainy season (the long rains) from March to May and a short rainy season (the short rains) in October and November (see, e.g., [21]). Irrigation farming of rice and horticulture has extensively been carried out in the area since the 1950s by utilizing water from two local rivers, the Thiba and Nyamindi, and the irrigation system is currently managed under a public mechanism, the Mwea Irrigation Scheme (MIS). Households in the area predominantly engage in crop farming, and other industries are minor in the region.

The present project is to increase the amount of available water for irrigation in the area. Although farming in Mwea has relied on irrigation water, it is carried out mostly only in the

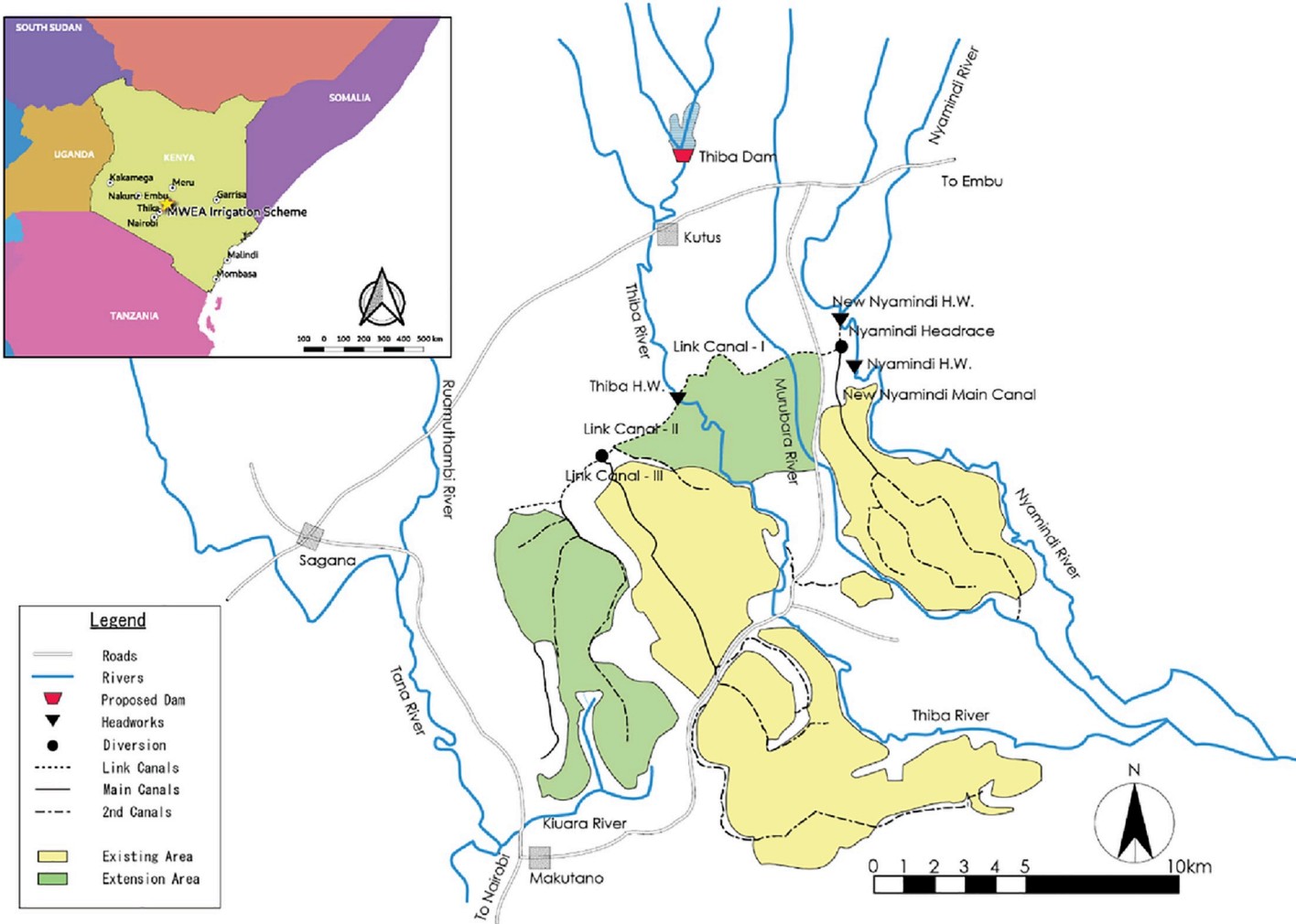

**Fig 1. Map of the Mwea irrigation scheme.**

two rainy seasons as irrigation water is limited and needs to be supplemented by rainwater. This general limitation of irrigation water, along with local awareness of profitability of rice farming and demographic pressures, is reflected in local interest in a new infrastructure project. With approximately 8,500ha of irrigated area and about 7,500 farming households, the MIS in its present form is by far the largest of the country's irrigation schemes and is where most rice production is undertaken in Kenya. Of the national irrigation schemes, the MIS accounts for about 80% of the total irrigated area and 90% of rice production [22,23]. Rice is the third most important cereal crop in Kenya after maize and wheat, and its consumption is rapidly growing. However, the vast majority of rice consumed in the country is imported, and the national government has set a long-term plan (the National Rice Development Strategy: NRDC) to increase rice production and thus reduce import dependency [24].

The irrigation development project began its implementation process in 2010 and construction is ongoing as of February 2019. In the planning stage, climate change adaptation was not considered an explicit objective of the project, and thus prior to this study, the project's effectiveness as a measure of climate change adaptation has not been assessed. Although it did not focus on the climate change implications, a feasibility study was made by JICA in 2009 (JICA

internal study for the Mwea Irrigation Development Project—detailed information in this study report is given in [25]).

Alongside the irrigation development project, a technical cooperation project jointly funded by the Government of Kenya and JICA, called the Rice-based and Market-oriented Agriculture Promotion (Rice MAPP) Project, was also conducted in the Mwea between January 2012 and January 2017 [26]. The Rice MAPP was not a project specifically addressing climate change adaptation but demonstrated a number of farming practices potentially useful for climate change adaptation, such as water-saving techniques, the cultivation of alternative crop varieties, and the extended storage of harvested rice. The Rice MAPP ran economic surveys on farmers' socioeconomic status and on the local rice market, and along with the outcomes of the JICA internal study, these data are used as a basis for our model calibrations.

## 2.2. Defining the problem

In this analysis, we evaluate the effectiveness of the irrigation development project on climate change adaptation defined as the difference in output quantities (e.g., yields) with and without the project under climate change relative to the difference of the same that would be expected in the absence of climate change–this follows the definition of adaptation benefits made by [18]. Note that in this analysis, we estimate both the general and overall benefits of irrigation under climate change, which are often the focus of discussion regarding the role of irrigation under climate change, and the reduced impacts of future climate change due to irrigation on farming, which should constitute the direct effects of adaptation. Non-irrigation uses of the water, such as power generation, drinking water supplies, and recreational use, are not considered in the project plan, and therefore their effects are not included in our analysis.

Economic metrics (income and others) are estimated by using a combination of simulation models, namely, climate, hydrological and yield forecasting models, which are soft-linked with each other. We do not build a single model including all these components but run computations by feeding the simulation outputs of each model to another–while this methodological choice of modeling does not capture the effects of how the behavior of local Kenyan farmers alters the climate and hydrological systems, the significance of such multi-directional interactions across systems is considered low in the context of the studied problem. Fig 2 illustrates the flow of our simulation analysis. It shows that the change in climate influences the water balance, and that both of them serve as input for yield forecasting, which determines economic outcomes. Water balance analysis and economic analysis are conducted by using the outputs of the three simulation models and also reflecting on the actual local conditions of cropping patterns, population, and so on.

Simulations are performed for a number of scenarios with varied climatic and socioeconomic parameters that represent the uncertainties for the Mwea (Table 1 for the list of uncertainties considered). The scenarios are treated equally without probabilistic weights, as they are from different models rather than being probabilistic scenarios predicted by a single model–this treatment of uncertainty is consistent with the RDM framework as mentioned above. It should be noted that the primary focus of our uncertainty analysis is to identify system behavior in the face of possible changes in key conditions, which local stakeholders are interested in. In other words, we are not concerned with the nature of the uncertainties. Hence, in the simulations we do not distinguish the treatment of uncertainties from different sources, such as natural randomness and incomplete knowledge, as long as the local stakeholders have no control over them.

To reflect local concerns about climate risks in the model simulations, we carried out our analysis in two stages–preliminary and main–holding a set of stakeholder interviews in

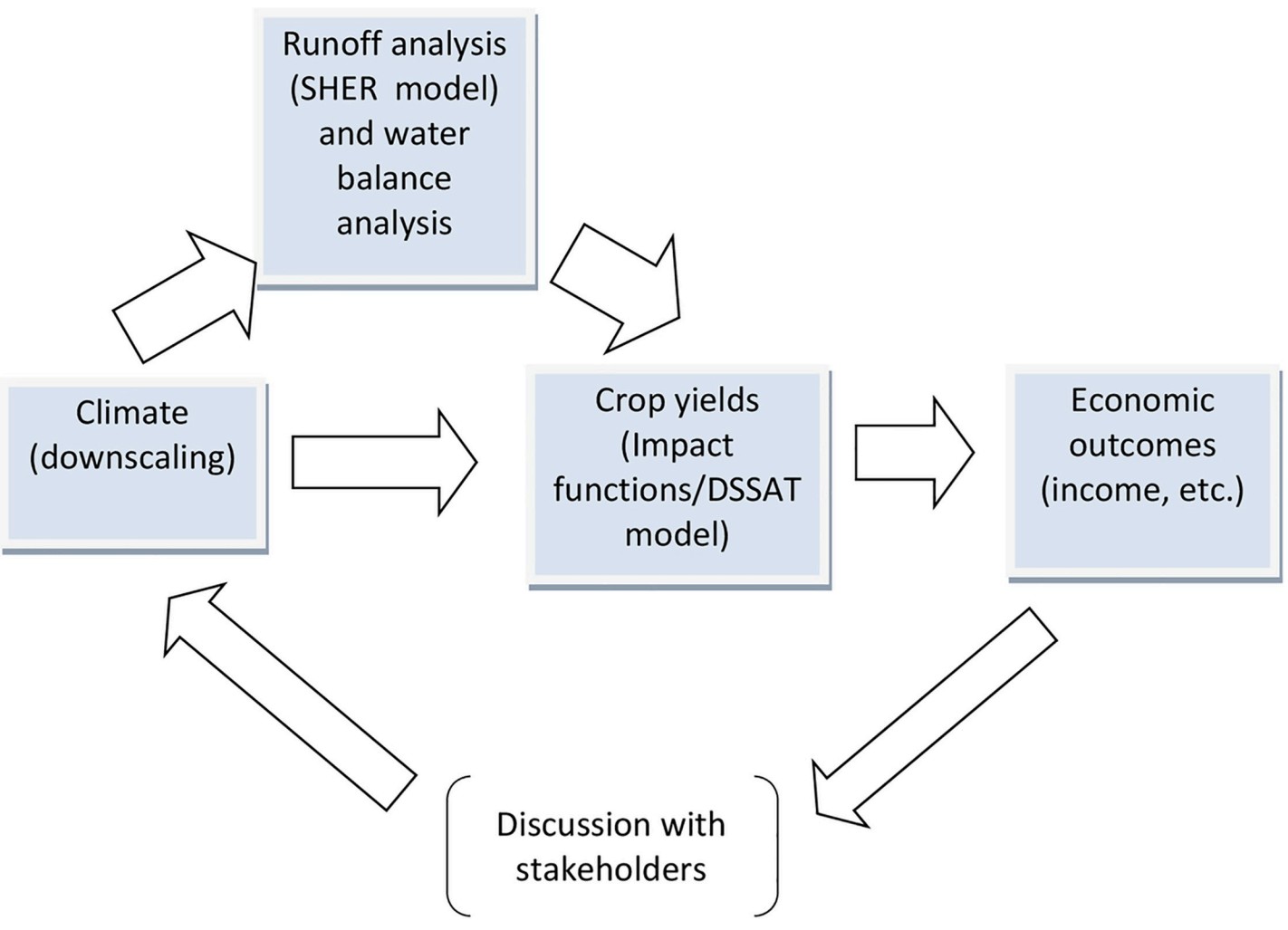

**Fig 2. Flow of the simulation analysis.**

between. The preliminary analysis (whose results are not reported in this paper) was comprised of a full set of model simulations, but its scenario and variable specifications were tentative and were made without local consultations. In May 2017 we held meetings with officials and representatives of the following organizations, presenting to them the preliminary results from our analyses, and obtaining feedback: the Mwea Irrigation Agricultural Development Centre (MIAD), the National Irrigation Board (NIB), the Irrigation Water Users Association (IWUA), the Ministry of Water and Irrigation, the Ministry of Agriculture, Livestock and Fisheries, and the Kenya Meteorological Department. We subsequently conducted the main analysis by reflecting their comments, and this paper reports on the results of the main analysis.

### 2.3. Development of climate scenarios

We develop scenarios of future climatic conditions in Mwea by using a simplified method of downscaling. The downscaling method we use for this analysis is a version of the delta change method as employed by Prudomme et al. [27]. In this method, changes of climatic variables (in our case, precipitation and temperature) are calculated as the differences in the output

Table 1. Types and number of scenarios (uncertainties) considered in the analysis.

| | Type of uncertainty | Number of scenarios | Note |
|---|---|---|---|
| Climate scenarios | RCPs ($CO_2$ concentration) | 4 | RCP 2.6, RCP 4.5, RCP 6.0, RCP 8.5 |
| | Temperature changes | 60 (LHS sampling) | Sampling range set by outputs of 14 GCMs |
| | Precipitation changes | | |
| | Seasonality change of precipitation | | |
| | (Subtotal) | (240) | |
| Socio-economic scenarios | Household number in MIS | 100 (LHS sampling) | In 2030 |
| | | | Upper bound: 47% increase |
| | | | Lower bound: no increase |
| | | | In 2050 |
| | | | Upper bound: 125% increase |
| | | | Lower bound: no increase |
| | Price of rice | | See S4 File for specifications |
| | | | Upper bound: no change |
| | | | Lower bound: 15% decrease |
| | Price of upland crops | | Upper bound: 10% increase |
| | | | Lower bound: 10% decrease |
| | Production cost | | Upper bound: 30% increase |
| | | | Lower bound: 30% decrease |
| | Discount rate | | Upper bound: 10%/yr |
| | | | Lower bound: 5%/yr |
| | | 24,000 | |

values of global circulation models (GCMs) for the baseline (current) and future periods on the model grid encompassing the target location. Then the estimated changes are added or multiplied to the observational weather levels at the baseline period. More precisely, future levels of climatic variables are computed according to the following formula for a climate variable $X$:

$$(1) \qquad \tilde{X}_{iy,im,j}^{GCM\_fut} = CF_{im} \cdot X_{iy,im,j}^{OBS\_hist} \qquad \text{(for precipitation)}$$

or,

$$(1') \qquad \tilde{X}_{iy,im,j}^{GCM\_fut} = CF_{im} + X_{iy,im,j}^{OBS\_hist} \qquad \text{(for temperature)}$$

Where: $iy$, $im$, and $j$ are indices for the year, the month and the day of the month, and a tilda (as in $\tilde{X}$) denotes the projected future value. A *CF* is the change factor, which is defined as the absolute (for temperature) or percentage (for precipitation) change in the means between the baseline and future time-slices.

The GCM data used for these estimations are obtained from the CMIP5 (Coupled Model Intercomparison Project Phase 5) database, which is publicly available (https://esgf-node.llnl.gov/projects/cmip5/) and contains simulation results of a number of GCMs used for the IPCC AR5 report. It would not have been appropriate, however, that we use data from all models indiscriminately for our purpose because some of these global models grossly misrepresent the specific climatic conditions of Kenya. Hence, for our dataset, we screen the available 47 GCMs in the dataset by applying the Interquartile Range Rule to temporal and spatial correlations and root mean square errors (RMSEs) of the model outputs for the present (evaluated for the area 10˚S-10˚N, 25˚E-50˚E). We select the outputs of 14 models that have passed this screening, each of which has datasets for different RCP (Representative Concentration Pathway) scenarios (which are comprised of RCP 2.6, RCP 4.5, RCP 6.0, and RCP 8.5: see [28]). Differences in RCP scenarios lead not only to different predicted future climatic conditions but also to

differences in the carbon dioxide fertilization effect, which influences crop yields. Note that from the standpoint of local stakeholders in Mwea, the global emission paths are basically fully given rather than subject to their control, and thus the simulation analysis does not distinguish the scenario differences of different GCMs and different RCPs.

In addition to the GCM data, observational weather data are needed for the application of the delta change method mentioned above. For Mwea, some observational local weather data exist but have frequent interruptions. Therefore, we primarily use the WFEDI (WATCH-Forcing-Data-ERA-Interim) reanalysis data ([29]: 0.5˚ x 0.5˚ resolution), a global dataset of model-interpolated observational weather data, for the delta-change adjustment of GCM-based values.

Since farming practices in Mwea are sensitive to the seasonality of precipitation, estimation of future changes in the seasonal patterns of rainfall (especially the onset of the rainy seasons) could greatly affect our results. We therefore consider future changes in rainfall seasonality by representing the seasonal variation in a functional form. Estimated future changes in monthly values of precipitation from each GCM are fitted to a harmonic function with one node, which is characterized by amplitude ($C_1$) and phase ($\varphi_1$) parameters as well as shifts in the annual average ($X_0$). The ranges in the levels of these parameters and of shifts in annual averages of temperature (more precisely, those of the daily maximum and minimum temperatures) are considered the ranges in our case sampling. As an alternative set, we also generate scenarios ignoring the change in precipitation seasonality. We do not conduct a fitting with a harmonic function for the seasonality of temperatures since seasonal variations of temperatures are relatively small and play in the context of Kenya a relatively minor role in farming.

The change factor $CF$ is computed according to the following formulae:

(2)        $CF_{im} = X_0 + C_1 \cdot \cos\left(\frac{2\pi \cdot im}{12} - \varphi_1\right)$        (for precipitation)

(2')        $CF_{im} = X_0$        (for temperature) where $im$ is the index for the month.

For each climate scenario, we set up a spatial dataset (with 0.1˚ x 0.1˚ grids for the region 0.9˚S-0.1˚S, 37.1˚E-37.6˚E) of the following weather variables for three time periods, the present/historical (average over the period 1991–2010), the reference year of 2030 (average over the period 2021–2040), and the reference year of 2050 (average over the period 2041–2060). The variables are precipitation, temperature (daily average, maximum and minimum), surface wind speed, relative humidity, shortwave radiation, and surface air pressure. The last four weather variables are necessary for calculating evapotranspiration (i.e., the reference evapotranspiration $ET_0$ by the FAO Penman-Monteith method), which is an input for the hydrological model described below. For these four variables, we use the values of present observational data (WFEDI reanalysis data) for all the three modeling periods by assuming that the levels will remain unchanged in the future.

Since one of the main objectives of our analysis is to assess the sensitivity of outcome variables to uncertainties of plausible climate conditions and parameter levels in the future, we do not use the above-described estimates of future climate from the 14 models directly but rather construct climate scenarios by using the following method of random case generation. We first identify the upper and lower bounds in the shifts in the levels of the weather variables among the constructed climate data of the 14 models and then preform a Latin Hypercube Sampling (LHS) from the identified ranges to generate a set of scenarios (60 scenarios).

## 2.4. Runoff analysis and water balance analysis

The amounts of available irrigation water in Mwea are modeled in a semi-empirical fashion by using a physical hydrological model that covers the catchment areas of Thiba and Nyamindi Rivers (no glaciers exist in the catchment areas of these rivers), and calibrating this with the

relationship between the constructed weather data from the WFDFI and the observational data of river flows at two monitoring stations in the Mwea area (Thiba and Rupingazi stations) during the period 1981 to 2010. The calibrated model is used to compute river flows and water distribution across the farming areas in the MIS, while applying allocation rules consistent with present farming practices and water rights under the MIS (details on this are discussed in [25]).

For the simulations of hydrological processes, we use the SHER (Similar Hydrologic Element Response) model, which is a hydrological model originally developed by Herath et al. [30] and applied in a wide range of evaluations of water resource related projects, including the estimations for the Kenyan National Water Master Plan 2030 published in 2014 (https://wasreb.go.ke/national-water-master-plan-2030/). The SHER model is a physical model that represents the hydrology of a watershed by a set of simple physical equations with parameters (porosity and conductivity of soil, and so on), whose levels are empirically set by using observational data and the results of laboratory testing. The model simulates hydrological processes by differentiating between recharging and discharging areas, the latter being the surroundings of Mwea in our case study, and the distinction of these two types of areas is made manually by the analyst. The version of SHER we use covers the catchment areas of Thiba and Nyamindi Rivers from Mount Kenya (upstream) to the Mwea area and is composed of sub-models of surface flows, subsurface flows, and aquifers. For parameterization, it utilizes publicly available datasets of soil type and land use: for soil type data, the KENSOTER (Kenya Soil and Terrain database, version 2.0) by the Kenya Soil Survey (KSS) and ISRIC, and for land use data the Africover database by the FAO. A detailed model description is given in the Water Master Plan document.

The available amounts of water for irrigation in Mwea are estimated as the differences in the modeled river flow rates at the three water intake points of the MIS (the New Nyamindi Headwork, Thiba Dam, and Thiba Headwork, all shown in Fig 1) and the amounts of the minimum required flows rates, i.e., the total amounts of the existing water rights in the areas downstream from the three points plus the minimum amounts of water flows necessary to keep the integrity of the river systems. Reflecting the local river conditions as identified by the JICA internal study, we set the minimum required flow rates at the New Nyamindi Headwork, Thiba dam and Thiba Headwork at 0.88, 0.98, and 1.86m$^3$/s, respectively. Precipitation and water flow data are available on a daily basis, but we only used monthly averages for analysis as our methodological approach does not yield reliable estimates of extreme weather values. However, since the target of our analysis is irrigation farming whose objective is to smooth out water inputs for crops, the omission of day-to-day fluctuations of precipitation does not greatly influence modeling results.

Collected water at the Headworks is distributed to farmlands through irrigation channels, which are subject to irrigation efficiency rates reflecting water leakage (See S3 File). We simulate water allocations across farmlands that are made proportional to the actual existing water rights, in other words, they may not be optimal distributions to maximize regional crop production (the possibility that a better water allocation might raise the farming productivity is not considered, given the reality of actual water management in the MIS). In the current farming practices of Mwea, a substantial number of farmers are utilizing irrigation water without official water rights (they are called "out growers" in the area). In the simulations, despite their informal status, we assume that these farmers will use water in the same way as the entitled farmers. Water supply affects yields and economic outcomes through influencing the sufficiency rate of water for crop farming, i.e., reduced water distributions lead to less than optimal water inputs for individual croplands and consequently result in a decline in yields. Seasonal water demand depends on the growth of cultivated crops and local cropping patterns.

## 2.5. Yield forecasting

For yield simulations, we construct impact functions of crop yields for changes in climatic variables from the baseline conditions of Mwea by approximating the outputs of DSSAT (Decision Support System for Agrotechnology Transfer) version 4.6 model simulations for Mwea farming into polynomial functions of temperature and water input. We use functional approximations of DSSAT rather than the model directly, for the reasons of favoring transparency in our modeling assumptions and also of managing the practical needs of computing numerous scenarios. The functions are built for rice (Basmati 370, the rice variety mainly grown in Mwea) from ordinary paddy production and ratoon (the second shoot from the base of a rice plant after cropping) production, and for five major upland crops (maize, tomato, green gram, French beans, and soybeans). Yield impacts are represented as the percentage change in yield from the baseline level, and functional forms are differentiated for rice production in the short and long rainy seasons. Specific forms of these functions are given in S2 File.

DSSAT is a software program consisting of 40 crop models [31] and is applied in a wide range of crop simulation analyses worldwide. For estimations of rice yields, among the models embodied in DSSAT, the CERES Rice Model is used for our analysis. Its crop models compute yields and other variables from the input parameters of weather conditions, soil type and conditions, and crop management. In our modeling context, the main advantage of using DSSAT over other crop simulation models is that it has already been applied in the analysis of Mwea rice farming [32], and thus information about parameter calibration is partly available. For simulations, we adopt the calibration (genetic coefficients) of [32] for Basmati 370 cultivation in Mwea under the soil conditions corresponding to those of the local soil type (Eutric Vertisol). Meanwhile, DSSAT does not have parameter sets for one of the upland crops we consider (green gram) and for the exact varieties of the other crops grown in Mwea. For these, we use the parameter sets of DSSAT for a similar crop (soybeans for green gram) or similar varieties (for the upland crops other than green gram) and adjust the yield output by using the actual yield data in Mwea surveyed by the JICA internal study. Also, ratoon production is not explicitly considered in the DSSAT framework, for our simulations of ratoon cultivation we used the DSSAT simulation results for ordinary paddy rice and applied to them the yield ratio of 0.45 in terms of ordinary paddy production and ratoon production, which is consistent with data of previous local surveys (shown in the JICA internal study).

By interpolating model outputs from DSSAT simulations, in the setting of Mwea we set up yield functions of the two most influential variables affecting yields, seasonal mean daily average temperature and water input. The effects of the latter are estimated by running the model numerous times with different levels of irrigation water input while the other conditions are held constant. For most of the other parameters for model calculation, such as the timing and amount of fertilizer input, we set the levels to achieve maximal production under most scenarios. Meanwhile, the setting of planting dates, which influences yield results, is made consistent with local practices, as described in the JICA internal study and the RiceMAPP study (i.e., the dates are not optimally set to maximize yield values in DSSAT).

Yields are estimated separately for each of the considered RCP scenarios (RCP 2.6, RCP 4.5, RCP 6.0, and RCP 8.5), to reflect different levels of atmospheric carbon dioxide concentrations across these scenarios and consequently the different extent of the carbon dioxide fertilization effect, which partly offsets yield losses from temperature increases (DSSAT can estimate the carbon dioxide fertilization effect). Note that the scale of the carbon dioxide fertilization effect in yield forecasting models is generally subject to large uncertainties, and for this reason, some modeling studies exclude this effect in simulations (see for example [33]). In this sense, our loss estimates could be seen as being relatively conservative (i.e., small). In parallel with yield

computations, we estimate water demand for crops (crop coefficients $K_c$ by the Penman-Monteith method) at each stage of plant growth based on FAO guidelines [34] and use this information for the water balance analysis discussed above (Section 2.4). S1 File shows our estimates of crop coefficients. Here again, despite the fact that DSSAT can compute water demand, we do not use DSSAT directly for that purpose in favor of the transparency of our simulation assumptions.

## 2.6. Economic analysis

Economic analysis is carried out for a hypothetical case without an irrigation dam ("donothing") and for four options of possible cropping patterns after the irrigation development project is completed, namely "RiceRice," "RiceUpland," "RiceRice+," and "RiceUpland+," which differ in crops grown in the long and short rainy seasons and in the adoption of improved farming practices and techniques proposed by the Rice MAPP project. These practices include the following: the Water Saving Rice Culture (WSRC), a set of water saving techniques for rice farming; the Improved Ratoon Production (IRaP), improved practices of water and fertilizer applications to raise yields of ratoon farming; the Warehouse Receipt System, the storage of harvested rice to sell it in the peak season of demand; and mechanization of farming. Based on the RiceMAPP data, we set the yield increase effect of WSRC and IRaP to be 13% and 17%, respectively, and the water saving effect of WSRC to be 20% of water input. The RiceMAPP findings are also utilized as the basis of our parameterization that mechanization reduces the production costs by 10%, water input by 30%, and the harvest loss by 6 percentage points. See Table 2 for a description of the options.

There are no data about the current scale of farming of upland crops in Mwea, and in our analysis, we assumed equal land areas allocated for each of the five major upland crops (maize, tomato, green gram, French beans, soybeans) considered in the options RiceUpland and RiceUpland+. In principle, farmers can maximize their profits by changing the shares of cultivated crops. For example, growing tomatoes is highly profitable in Mwea, and thus a shift from rice farming to tomato farming should generally raise farmers' income. However, a

**Table 2. Options of cropping patterns and improved farming techniques with and without the irrigation development project.**

| Option name | Cropping patterns | Improved farming practices |
|---|---|---|
| No irrigation development project (donothing) | SR + SRR (in part SR + LR) | |
| With irrigation development project | | |
| RiceRice | SR + LR | |
| RiceUpland | SR + SRR + LRU | |
| RiceRice+ | SR + LR | WSRC + IRaP + WRS + mechanization |
| RiceUpland+ | SR + SRR + LRU | WSRC + IRaP + WRS + mechanization |

SR: Paddy rice cultivation in the short rainy season.

SRR: Ratoon rice cultivation after the short rainy season.

LR: Paddy rice cultivation in the long rainy season.

LRU: Cultivation of paddy rice and upland crops and in the long rainy season.

WSRC: Water Saving Rice Culture.

IRaP: Improved Ratoon Production.

WRS: Warehouse Receipts System.

number of determinants other than profit exist in the real-world context of local farming—for example, tomatoes are relatively easily perishable, and thus in the weak logistic system of Mwea a large quantity of tomatoes cannot be reliably delivered to major markets, not to mention the general high revenue volatility resulting from the farming of fresh vegetables. For this reason, while in our analysis we contrast scenarios of enhanced rice farming and farming emphasizing upland crops, we do not perform a detailed analysis of the varied compositions of cultivated crops in this study.

We estimate indices of economic outcomes for combinations of climate and socioeconomic scenarios listed in Table 1. For the socioeconomic scenarios, we generate 100 sets of these by an LHS method, which randomly selects combinations of the values of socioeconomic parameters whose possible levels are assumed to be distributed uniformly between given upper and lower bounds. We computed LHS using the pyDOE package of Python.

Among the parameters of socioeconomic scenarios, the size of population in Mwea (MIS) is a parameter affecting the household income. We set the baseline number of households at 7,453 according to a survey made in 2016 by the Rice MAPP project [26]. In the years 2030 and 2050 we consider the lower bound of population to be the current level (given the water limitations in Mwea). The upper bound is set to be consistent with the estimates of UN projections of rural population in Kenya, i.e., a 47% increase in 2030 and a 125% increase in 2050 (Table 1), as there are no population projections specific to the Mwea region. With the current irrigated area of 8,362ha (it will remain the same after the irrigation development project), this means that each farming household will have about 1ha, or 2.5 acres, with no population increase, and half that area with a doubled population. As identified by the survey carried out for the JICA internal study, although farmlands are not private properties in an official sense, they are mostly transferred to junior family members upon a scheme commissioner's consent when the head of an MIS household dies. Given the high population growth rates of Kenya including the Mwea area, it is likely that farmlands in Mwea will remain settled by family members of the existing farmers in the future (i.e., an inflow of new farmers from outside is unlikely).

For the evaluation of farmers' income, we use wholesale crop prices estimated as in S4 File. Meanwhile, for cost-benefit evaluation of the project, we employ an alternative set of crop prices for evaluation based on the world prices of crops in the World Bank Commodities Price Forecast. The baseline crop prices are set based on the data from the following three sources: the Rice MAPP survey in 2016 [26], the National Farmers Information Service (NAFIS: www.nafis.go.ke) by the Ministry of Agriculture, Livestock and Fisheries, and the JICA internal study. We then estimate the 2030 prices by applying the growth rates of the world commodity prices forecast by World Bank Commodities Price Forecast (as of October 26, 2017, http://pubdocs.worldbank.org/en/678421508960789762/CMO-October-2017-Forecasts.pdf, last accessed 28 January 2018) to these baseline prices, and we assume no systematic trend of prices from 2030 to 2050. The base year of currency units is the year 2010. As crop farming is the dominant source of income for the farmers in the MIS (which is identified by, for example, the Rice MAPP survey: [26]), farmers' income is evaluated as the sales revenues from crops at wholesale prices minus the production costs, whose information was collected through local surveys performed by the JICA internal study.

Along with the above analysis, we also compute the conventional metrics of cost-benefit analysis such as the net present value (NPV) for the irrigation development project, reflecting climate change and uncertainties. To this end, multiple possibilities of the social discount rate (values randomly chosen between 5 percent/yr and 10 percent/yr) are also considered in the LHS sampling.

## 3. Results

### 3.1. Reconstruction of local climatic and hydrological patterns

Fig 3 shows our estimated changes in annual average temperature and precipitation for 2030 and 2050 from the baseline levels under different RCP scenarios, represented in a range (i.e., the highest and lowest values of model-derived estimates). The graphs exhibit a clear tendency for increasing temperatures over the years, by around 1˚C in 2030 and somewhat greater than that level in 2050. On the other hand, the future changes in precipitation are unclear even in sign, although the ranges tend to be skewed in the positive direction (i.e., greater amounts of precipitation).

Fig 4 shows the mean river flow rates at the New Nyamindi Headwork under different climate scenarios, represented as scatterplots of mean flow levels in four seasons (the upper graphs are for the long and short rainy seasons (the long rains and the short rains), and the lower graphs are for the dry seasons). The figure presents two sets of results, one for 2030 (left graphs) and the other for 2050 (right graphs). The black points correspond to the baseline levels, and the blue and yellow circles represent the estimates without and with seasonality adjustments of precipitation by approximation with a harmonic function. The graphs indicate a general increase in river flows in Mar-Apr-May (the long rainy season) and Jan-Feb (one of the dry seasons). Meanwhile, the trends in water quantities in the other seasons are unclear. Also, results for 2030 and 2050 are very similar, except that the latter exhibit a somewhat greater variance than the former. Adjustments of seasonality with a harmonic function generally yields greater river flows in Mar-Apr-May and Jan-Feb.

### 3.2. Project effectiveness under climate change (yield and economic impacts)

Fig 5 shows the box plots of simulation results of farmers' annual income and the total rice production of Mwea (MIS) for the four options in the irrigation development project and for the case without the irrigation development project ("donothing"). Each plot represents income level under one of 24,000 scenarios in Fig 5(A), or rice yield under one of 240 scenarios for Fig 5(B) (see also Table 1), in other words, their distributions represent uncertainties. The ends and middle lines of the boxes correspond to the quantiles (i.e., the middle line represents the median), and the whiskers represent the 1.5 interquartile ranges (IQRs). The "baselines" represent the present levels, which are reconstructed values from the yield forecasting and other models assuming the baseline climate conditions (i.e., not the averages of actual levels for any specific period, although they are indeed approximately at the same levels as the current levels found by the local surveys mentioned earlier).

The figure shows that estimated income levels for the four options with the irrigation development project are generally higher than those without the irrigation development project, although some overlaps in the range of boxes and whiskers exist. Adoption of improved farming techniques also leads to generally higher levels of income relative to the cases without them (identified through comparison between RiceRice and RiceRice+, and comparison between RiceUpland and RiceUpland+), while the rice yields for the options involving double-cropping of rice (RiceRice and RiceRice+) naturally exhibit higher levels of rice production than those with the farming of upland crops (RiceUpland and RiceUpland+). Without the irrigation development project ("donothing"), farmers' income is reduced in the future periods in all scenarios, while the total rice production generally turns negative from 2030 to 2050. For the four options with the irrigation development project (RiceRice, RiceUpland, RiceRice+, RiceUpland+), results for the reference years 2030 and 2050 are mostly similar, except that the

(a) Change of annual temperature relative to the baseline (1991-2010 average)

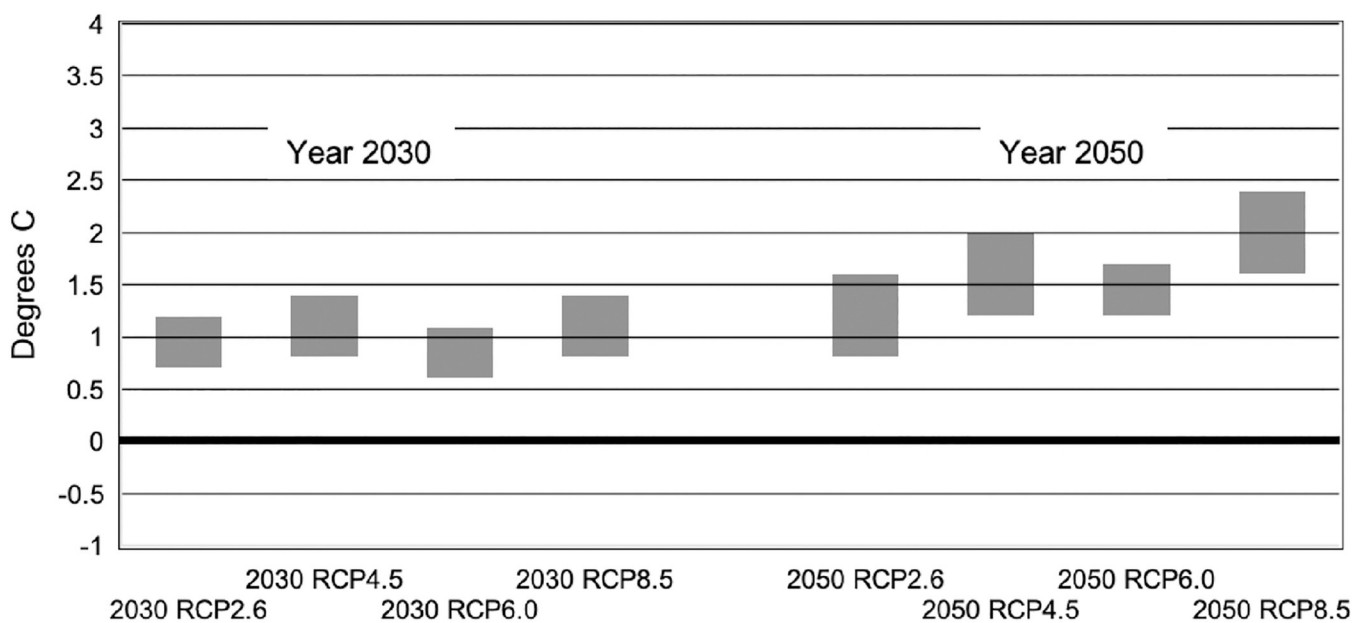

(b) Change of annual precipitation relative to the baseline (1991-2010 average)

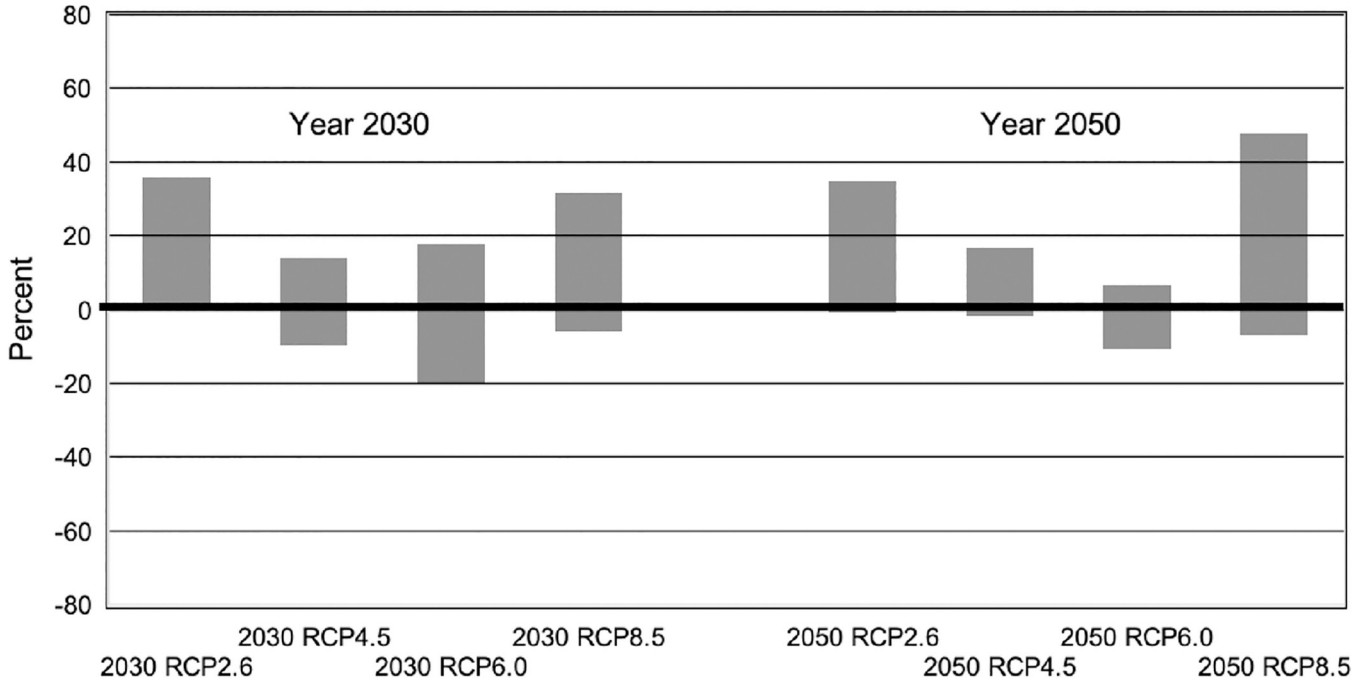

**Fig 3. Future changes in annual temperature and precipitation in Mwea estimated by the delta change method using 14 GCM data (represented in ranges).**

(a) Rainy seasons for the years 2030 (left) and 2050 (right). The vertical axis is for the long rainy season of Mar-Apr-May (the long rains), and the horizontal axis is for the short rainy season of Oct-Nov-Dec (the short rains).

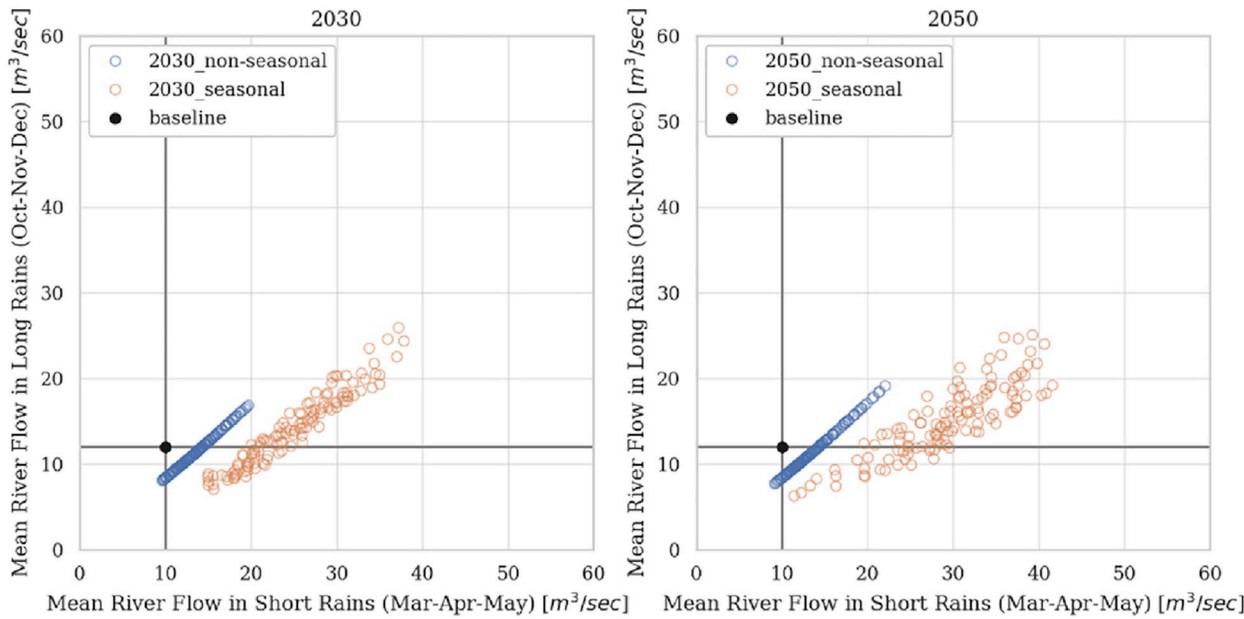

(b) Dry seasons for the years 2030 (left) and 2050 (right). The vertical axis is for Jan-Feb, and the horizontal axis is for Jun-Jul-Aug.

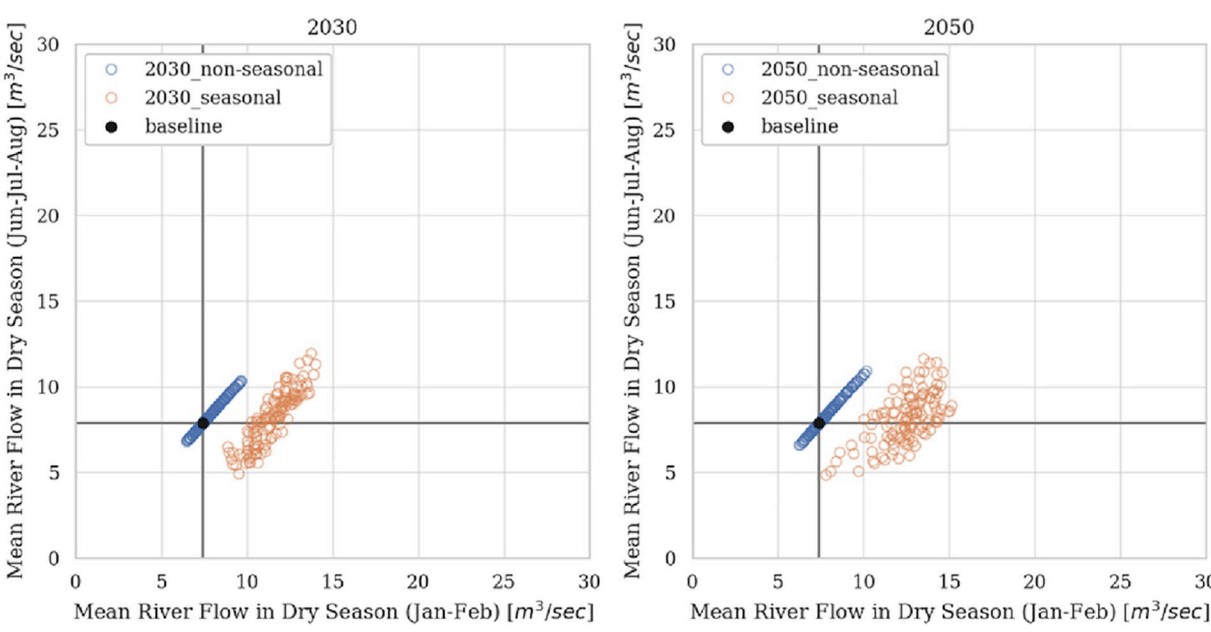

**Fig 4.** Estimated mean river flows at the New Nyamindi Headwork for 2030 (left graphs) and 2050 (right graphs) for the short and long rainy seasons (a) and the dry seasons (b).

variance of results becomes somewhat magnified from 2030 to 2050 due to greater uncertainties of socioeconomic scenarios in a more distant future.

The effects of climate change are made more visible in Fig 6, which isolates the relative changes of rice yields (Graph a) and average household income (Graph b) in Mwea from the baseline levels that assume no climate change, corresponding to the adaptation benefits as defined by [18]. The results are shown as percentage changes from the baselines. For both graphs, the dotted horizontal line corresponds to zero change (no effects of climate change), and the boxes represent changes from the baseline under different climate and socioeconomic scenarios.

Overall, the plots indicate great uncertainties about climate change impacts, with many areas both below and above zero for all the five sets of results. However, especially for 2050, the general tendency of negative yield changes under climate change is clear (recognizable by, for example, the fact that the median is negative)–this is a noteworthy result given that the predicted trends of precipitation are ambiguous, in other words, elevated temperatures alone could cause yield losses. In 2050, the yield and income losses caused by climate change are most significant for the case without the irrigation development project (donothing), while this tendency is not clear for the year 2030 (in fact, a majority of the donothing results for 2030 exhibit positive changes in yield and income). At least for 2050, relative to the donothing results, the distributions for the other four options are generally higher both in yield and income, and many scenarios of the four options with irrigation development exhibit positive income changes even under climate change, reflecting improvement and diversification of crop farming. For example, at the median and for the year 2050, the income loss of 6% for the no-project (donothing) case is flipped to become positive in the other four cases with the project, while the yield loss of 4% for no project is also slightly reduced with the four project options.

These results are an indication that the irrigation development project mitigates the negative effects of climate change, in other words, it serves as an effective means of climate change adaptation. It is also noticeable in Fig 6 that the spread of boxes is greater in 2050 than in 2030 –this corresponds to the greater uncertainty of climate change in the former year than in the latter year. The worst possible outcome for the donothing case for 2050 is greater than 50% of income loss which would be mitigated by the irrigation development project.

Fig 7 presents box plots on the project's NPV inclusive and exclusive of climate change effects ("with CC" and "no CC," respectively) for the four project options. The results are consistent with those presented on Figs 5 and 6, and the project benefits are generally magnified when climate change is taken into consideration.

## 4. Discussion

Our simulation results show that despite the uncertainties in precipitation trends, the higher temperatures resulting from climate change will have a general tendency to reduce farmers' income due to lower crop yields, and that irrigation development will mitigate that income loss, i.e., it will likely function as a means of climate change adaptation. Here, the adaptation benefits are estimated as an effect of reduced climate change damage isolated from the general benefit of irrigation, in consistent with the formulation of [18]. This adaptation benefit of irrigation development on income is reflected in the total project benefit (NPV) as well, although the adaptation effects on the latter tend to be very small because the effects on income become

## (a) Farmers' annual income

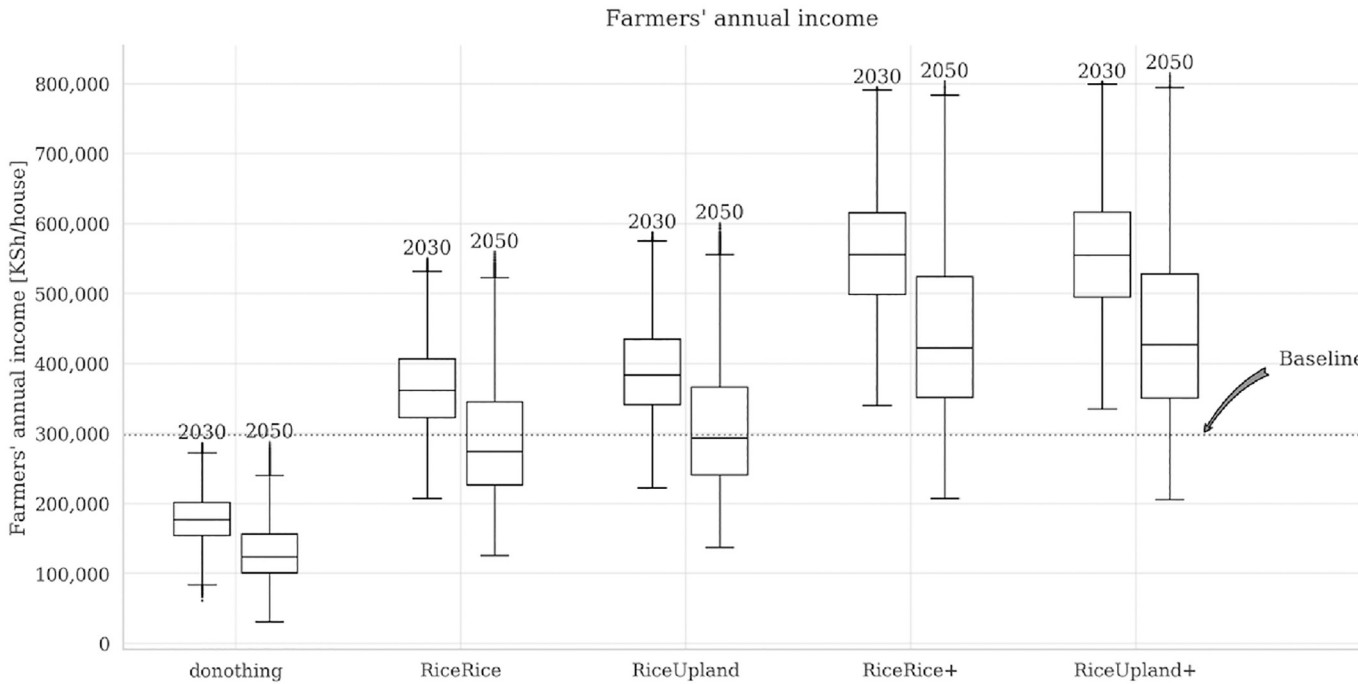

## (b) Total rice production of the Mwea Irrigation Scheme (MIS)

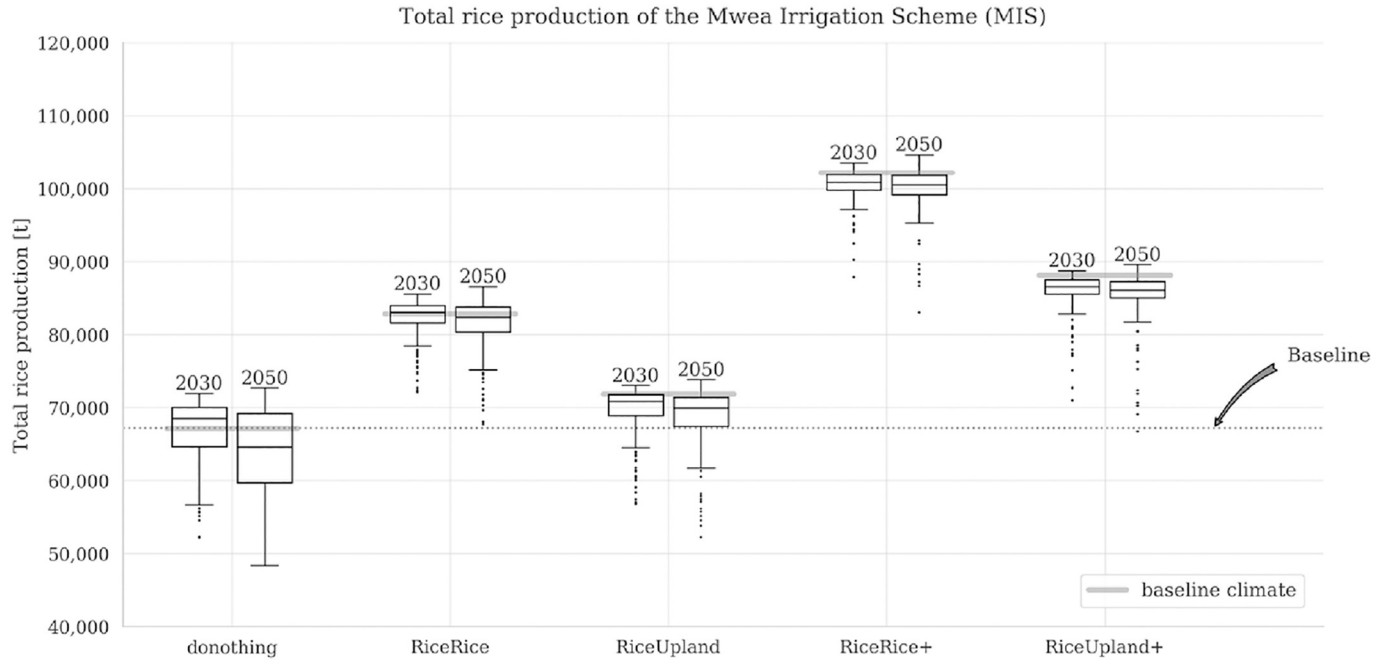

**Fig 5. Simulation results of farmers' annual income and the total rice production of the Mwea Irrigation Scheme (MIS).**

## (a) Rice yield

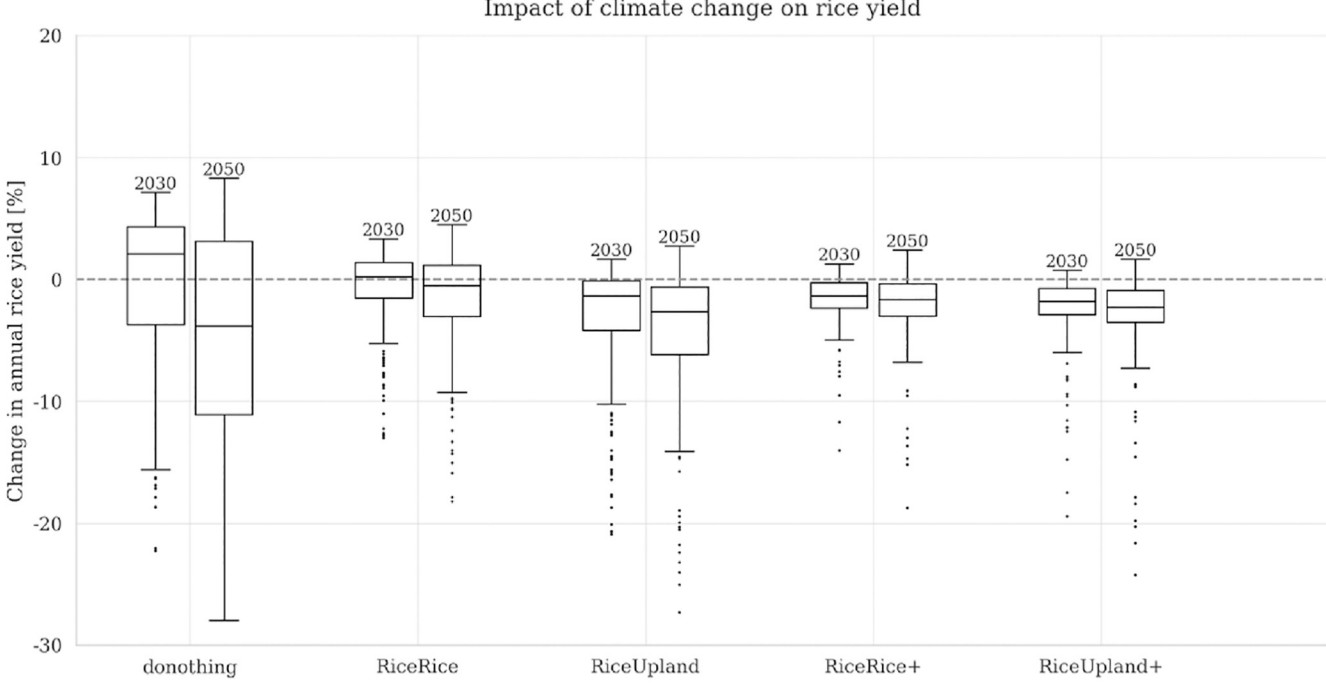

## (b) Average household income

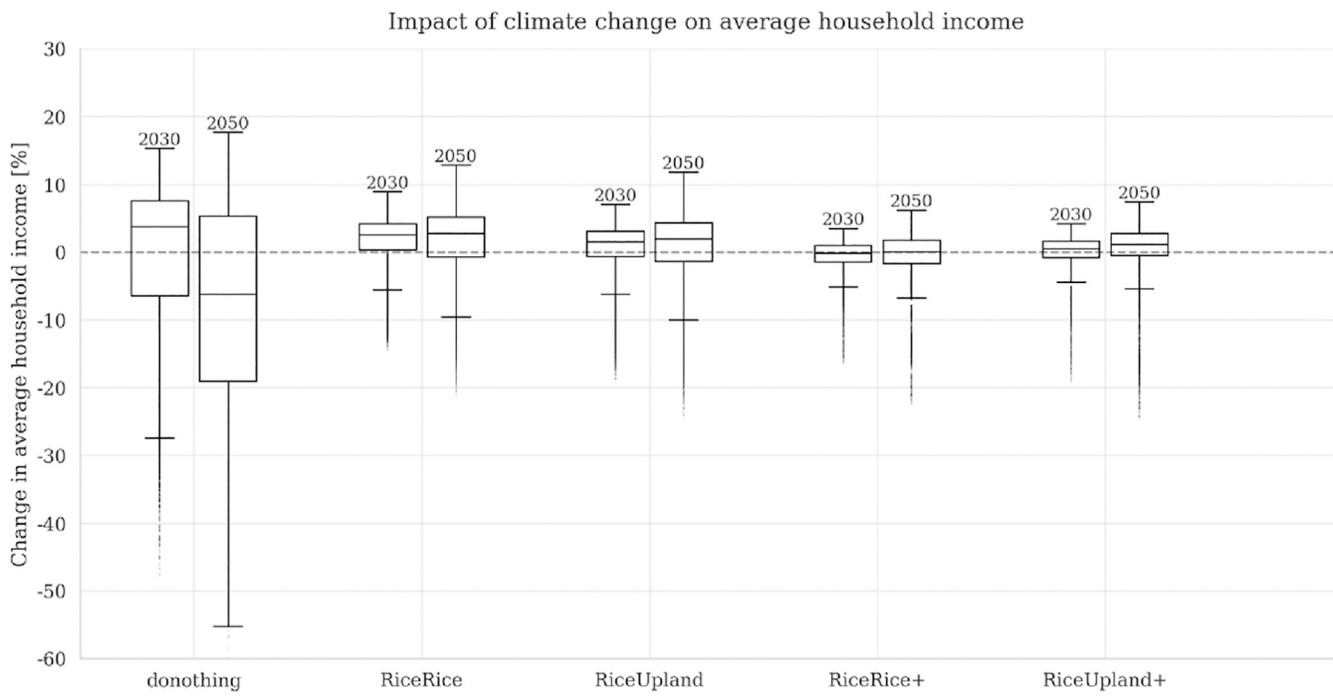

**Fig 6. Impact of climate change on the rice yield and average household income in the Mwea Irrigation Scheme (MIS) (percentage changes under climate change relative to the levels without climate change).**

progressively strong towards the end of the project term and are thus heavily time-discounted in the NPV calculation. The results also indicate that irrigation could mitigate the worst possible outcomes of yield and income loss under climate change that would be realized without such irrigation development projects in combination with the worst possible socioeconomic conditions. A further important insight obtained from the results is that the effects of non-irrigation measures, such as adoption of water-saving techniques, on income enhancement are substantial.

Our basic finding about climate change impact on yields, i.e., the trend of significant negative effects of climate change, is broadly consistent with the findings of existing studies dealing with climate change impacts on agriculture in Africa, such as [5,10,12]. Consistency with the existing studies regarding yield responses is not surprising because it is based on one of the widely-used model frameworks of yield forecasting, as is the case for [35] of African rice farming. Comparison of results at a finer level is not possible because our estimations are broadly based on the present local cropping conditions that are not necessarily optimized to the current climate, while the other modeling studies assume representative farming conditions without local peculiarities. Meanwhile, our results are also in line with the conclusion of [36], in that temperature changes are more important for crop yields than precipitation changes, and thus that despite the inconclusiveness of precipitation trends in the target area under climate change, a clear tendency of negative effects of climate change shows up because of the unambiguously increasing trends of temperature alone.

At the same time, however, some of our findings depart from those of existing studies as well. First, our analysis is not a mere assessment of climate change impacts but isolates the

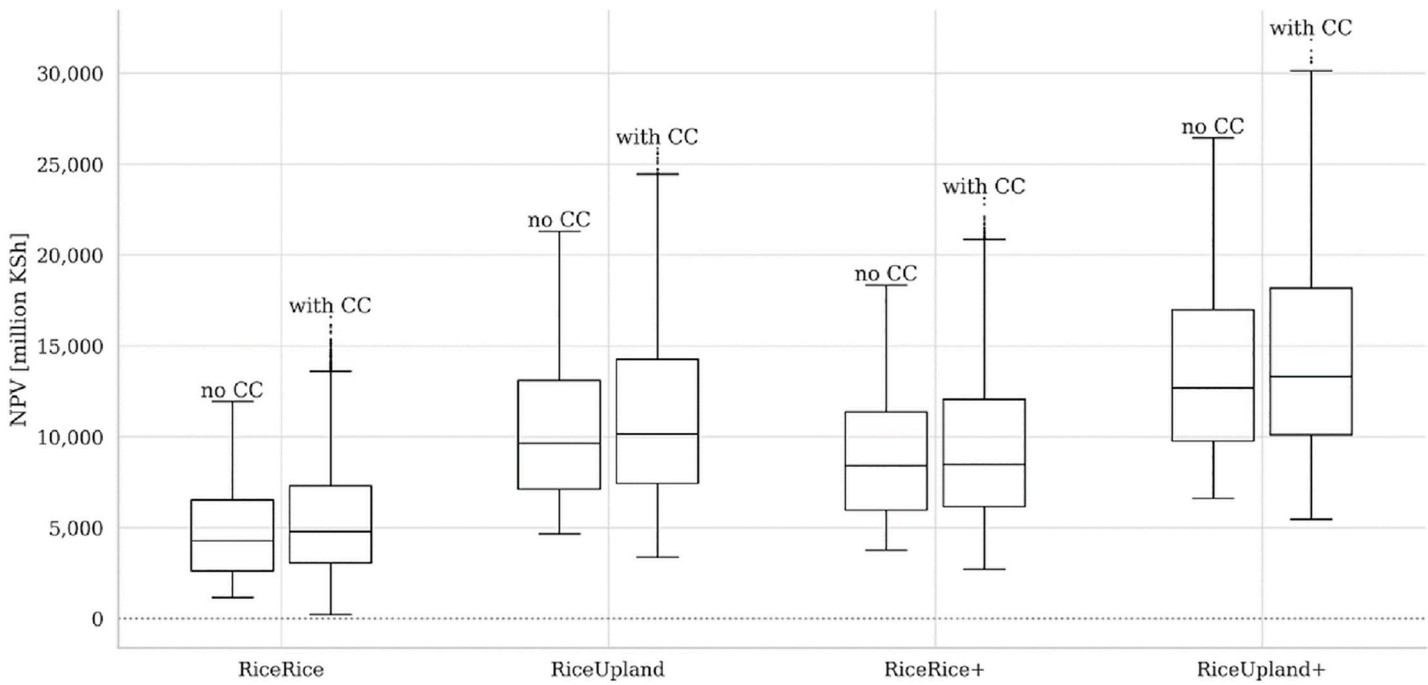

**Fig 7. NPV of the Mwea Irrigation Development Project for the four different project options, inclusive and exclusive of the effects of climate change ("with CC" and "no CC," respectively).**

direct effects of reduced climate change damage due to irrigation development and other measures of interventions. Also, as our study reflects various local conditions and constraints such as those of local weather, water allocation and economic factors, the estimated impacts show a wide range of levels, some of which are differing in sign. Positive impacts of irrigation development are greatest for the worst-possible cases of climate change damages.

Unlike macro-level studies dealing with climate change impact on agriculture in Africa, our case-based approach can reflect realistic local institutional arrangements and circumstances, e.g., the local needs for the support of an increasing number of farmers in response to rapid rural population growth, not to mention that it can also incorporate detailed local characteristics of climate, hydrology, agronomy, and socioeconomic conditions.

## 5. Conclusion

Various types of public interventions, such as those of irrigation, are potentially beneficial for people's economic conditions under climate change and are hence proposed to be the means of climate change adaptation. But the current policy discussions of climate change adaptation often treat the concept of adaptation loosely and do not refer to rigorous metrics of adaptation effectiveness that isolate the reduced impact of future climate change due to the intervention from the general project benefits.

In response to such lack of evaluation of adaptation effectiveness, by using a combination of simulation models, we conducted a case study of a Kenyan irrigation development project and quantified such isolated effects of climate change adaptation. Besides the general positive effects of the irrigation development project on local farming under future climate change, the simulations identified the benefits of irrigation development strictly attributable to climate change adaptation, i.e., climate change has the tendency to reduce crop yields and farmers' income, and an increase of water availability by irrigation development mitigates such negative impacts. Our study also found the range of possible negative economic outcomes resulting from uncertainties is wider for the cases without irrigation than with irrigation, indicating, among other things, the very large potential yield losses that could occur under climate change in the worst cases without new irrigation infrastructure. While our case study was made in the context of a specific locality and project, it offers some general hints that irrigation could bring some adaptation benefits for farming in the face of climate change, and that it may also strengthen the local capacity to deal with the worst possible outcomes of climate change.

We used standard, publicly-available data and models for the simulation of climatic conditions, yields and economic outcomes. This means that our assessment approach could be applied to many other cases of irrigation development, including those in relatively data-scarce developing countries. Further case- or project-based insights of climate change adaptation are particularly needed in the areas of climate finance and project appraisal of public investments.

## Supporting information

**S1 File. Estimated crop coefficients.**
(DOCX)

**S2 File. Yield functions.**
(DOCX)

**S3 File. Estimated irrigation efficiency levels in the MIS (adapted from JICA and Nippon Koei 2018).**
(DOCX)

**S4 File. Estimated wholesale prices of crops.**
(DOCX)

## Acknowledgments

We benefited from discussions with and inputs by the following people: Ichiro Adachi, Kotaro Taniguchi, Hiroshi Takeuchi, Daigo Makihara, Yuji Masutomi, Kiyoshi Takahashi, Koji Dairaku, Wycliff Nyang'au, Martin Gómez-Garcia, Julie Rozenberg, Laura Bonzanigo, Stephane Hallegatte, and Marianne Fay. Seminar and meeting participants at the following venues and institutions provided us with valuable insights: JICA, JICA-RI, the National Institute for Environmental Studies, the JpGU Meeting 2018, the 2019 AGU Fall Meeting, and the Kenyan institutions mentioned in the main text.

## Author Contributions

**Conceptualization:** Daiju Narita, Ichiro Sato.

**Formal analysis:** Daikichi Ogawada, Akiko Matsumura.

**Methodology:** Daiju Narita, Ichiro Sato, Daikichi Ogawada, Akiko Matsumura.

**Writing – original draft:** Daiju Narita.

**Writing – review & editing:** Daiju Narita, Ichiro Sato, Daikichi Ogawada, Akiko Matsumura.

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
