## [Decision Letter · Decision Letter 0]

8 Sep 2020

PONE-D-20-19964

Integrative Economic Evaluation of an Infrastructure Project as a Measure for Climate Change Adaptation: A Case Study of Irrigation Development in Kenya

PLOS ONE

Dear Dr. Narita,

Thank you for submitting your manuscript to PLOS ONE. After careful consideration, we feel that it has merit but does not fully meet PLOS ONE’s publication criteria as it currently stands. Therefore, we invite you to submit a revised version of the manuscript that addresses the points raised during the review process.

There is need for improvement on the framing of the research question from the introduction and in the discussion which are important in understanding the role of irrigation in climate change adaptation. In addition, our reviewers suggest that you have a clear separation of results and methodology for better reproducibility, improve on some figures and discuss the context of the study for potential transferability of the findings to other areas, either in Kenya of in the region.

We look forward to receiving your revised manuscript.

Kind regards,

Abel Chemura

Academic Editor

PLOS ONE

Journal Requirements:

We note that one or more of the authors are employed by a commercial company: Nippon Koei Co., Ltd,.

2.1. Please provide an amended Funding Statement declaring this commercial affiliation, as well as a statement regarding the Role of Funders in your study. If the funding organization did not play a role in the study design, data collection and analysis, decision to publish, or preparation of the manuscript and only provided financial support in the form of authors' salaries and/or research materials, please review your statements relating to the author contributions, and ensure you have specifically and accurately indicated the role(s) that these authors had in your study. You can update author roles in the Author Contributions section of the online submission form.

2.2. Please also provide an updated Competing Interests Statement declaring this commercial affiliation along with any other relevant declarations relating to employment, consultancy, patents, products in development, or marketed products, etc.  

4. We note that Figure 1 in your submission contain map images which may be copyrighted. All PLOS content is published under the Creative Commons Attribution License (CC BY 4.0), which means that the manuscript, images, and Supporting Information files will be freely available online, and any third party is permitted to access, download, copy, distribute, and use these materials in any way, even commercially, with proper attribution. For these reasons, we cannot publish previously copyrighted maps or satellite images created using proprietary data, such as Google software (Google Maps, Street View, and Earth). For more information, see our copyright guidelines: http://journals.plos.org/plosone/s/licenses-and-copyright.

4.1.    You may seek permission from the original copyright holder of Figure 1 to publish the content specifically under the CC BY 4.0 license. 

4.2.    If you are unable to obtain permission from the original copyright holder to publish these figures under the CC BY 4.0 license or if the copyright holder’s requirements are incompatible with the CC BY 4.0 license, please either i) remove the figure or ii) supply a replacement figure that complies with the CC BY 4.0 license. Please check copyright information on all replacement figures and update the figure caption with source information. If applicable, please specify in the figure caption text when a figure is similar but not identical to the original image and is therefore for illustrative purposes only.

Reviewers' comments:

Reviewer's Responses to Questions

**Comments to the Author**

1. Is the manuscript technically sound, and do the data support the conclusions?

Reviewer #1: Partly

Reviewer #2: Partly

2. Has the statistical analysis been performed appropriately and rigorously? 

Reviewer #1: Yes

Reviewer #2: Yes

3. Have the authors made all data underlying the findings in their manuscript fully available?

Reviewer #1: Yes

Reviewer #2: Yes

4. Is the manuscript presented in an intelligible fashion and written in standard English?

Reviewer #1: Yes

Reviewer #2: No

5. Review Comments to the Author

Reviewer #1: This compelling article models the agricultural and economic outcomes (yields and incomes, respectively) in 2030 and 2050, with and without use of irrigation, as climate change proceeds in Kenya’s Mwea area. The authors’ stated goal is to assess the extent to which a specific public irrigation project can serve as a form of climate change adaptation.

Some areas for improvement are identified below.

Although included in the abstract and at various points in the paper, “climate finance” is not clearly defined.

The paper’s contribution could be more clearly explained within the introduction:

• First, I find your clear descriptions of models (climatic, hydrological, agricultural, and economic) to be a strong point of the paper. It seems others might want to replicate this methodological approach. Might you choose to explicitly present this feature as a selling point of your paper somewhere in the introduction section, perhaps in the paragraph beginning “Our study aims…” (p3)?

• Secondly, the sentence beginning “However, to the authors’ knowledge…” (p3) is very complex, such that it makes it difficult to discern the paper’s unique contribution. Is it the economic appraisal of irrigation as a climate change adaptation strategy? I believe others have done that already. (For example, Cunha et al. 2015 model agricultural incomes in Brazil, and Finger et al. 2010 model economic outcomes in Switzerland.) If the paper’s contribution is that it is the first to model the economic aspect *of a specific irrigation project* as a climate change adaptation strategy, this could be clarified with a more direct/simplified sentence. I’d also like a bit more information about how your paper differs from those process-based and statistical models you mention.

Based on the introduction section, I expected the paper to discuss economic outcomes on a project-level scale; instead, the economic results were presented at the scale of household income. If I’ve correctly understood that project-level economic outcomes are one of the paper’s main contributions to the literature, I suggest creating some version of Appendix 5 as a main figure within the paper and discussing it in detail. Appendix 5 isn’t quite right for this purpose, though, as it appears to subtract the NPV that doesn’t account for climate change from the NPVs that account for climate change. Have you presented the NPVs under climate change anywhere? Phrasing like “project appraisal of public investments” (p15) led me to expect this as a principal finding.

I do not understand the “preliminary” and “main” stages of analysis described on p6 and in Figure 3. Did you only run the models for cropping strategies of interest? Clarification would be useful.

How did you select the four RCP scenarios (p7)? How many RCP scenarios exist?

For the economic analysis, could you justify your apparent assumption that the UN’s projected % increases in the rural population will be equal to the % increase in the farming population (and, therefore, the amount of land available to each farming household)?

I suggest improving the presentation of results as follows:

• The project area map (Figure 1) would benefit from a scale.

• In Figures 4a and 4b, it would be useful to darken the baseline. It would also be useful to make it clearer that the results are not temporal by clearly differentiating the 2030 values from the 2050 values. My first inclination was to read the bars from left to right, as if they progressed over time.

• For Figure 5, it is not clearly explained that the rainy seasons are together in one panel, and the dry seasons are together in the other panel. Might you briefly explain the benefit of grouping the seasons in this way? It would also be useful to write out “long rains” and “short rains” on the axis labels.

• For Figure 6, it is not clearly specified what the baseline represents. Current annual income and rice production? With data from which year?

Smaller points of clarification:

In the first introductory paragraph, it is unclear whether the UNEP-DTU estimates are the costs necessary to fully mitigate losses or are actual predicted expenditures in developing countries.

On the bottom of p5, the authors describe feeding simulation outputs of each model to another rather than building a single model. Are there downsides to this approach? Is this common practice (citations)?

On p6, when describing the RDM framework, it would be useful to briefly summarize the framework again rather than referring the reader back to the introduction section.

Might something like “no irrigation” be a clearer scenario descriptor than “do nothing”?

Reviewer #2: The manuscript deals with a very important subject involving climate change adaptation which is a contemporary area of research. Irrigation as climate change adaptation strategy is a fundamental subject and finding of the study. Though comprehensive, appropriate and relevant data have been collected and analysed, the authors need to improve a number of things in the manuscript including the title, explicit statement on objective of the study, clear separation of results and methodology so that comprehensive presentation of results and discussion are consolidated in one section. The methodology is not reported in past tense to indicate that it is a report of how the research was pursued to generate the results presented. Instead it is presented in present and continuous tenses as if it is not a report of how the research was pursued. The use of ""We" is also monotonous and is not encouraged. The conclusion need to be revisited and made more explicit in addressing the objective of the study. Generally the manuscript requires elaborate editing and synthesis. The details of the review are marked on the pdf format of the paper attached.

6. PLOS authors have the option to publish the peer review history of their article (what does this mean?). If published, this will include your full peer review and any attached files.

Reviewer #1: No

Reviewer #2: No

---

## [Author Response · Author response to Decision Letter 0]

1 Nov 2020

We gave our responses in the cover letter.

---

## [Editor Report · Decision Letter 1]

11 Nov 2020

PONE-D-20-19964R1

Integrating Economic Measures of Adaptation Effectiveness into Climate Change Interventions: A Case Study of Irrigation Development in Mwea, Kenya

PLOS ONE

Dear Dr. Narita,

Thank you for submitting your manuscript to PLOS ONE. After careful consideration, we feel that it has merit but does not fully meet PLOS ONE’s publication criteria as it currently stands. Therefore, we invite you to submit a revised version of the manuscript that addresses the points raised during the review process.

There has been some improvement from the initial submission and the submission should be re-structured into a proper research paper taking into account the suggestions on formatting, referencing style and presentation. A separate discussion section can also improve the synthesis of the results and their comparison with similar studies.

We look forward to receiving your revised manuscript.

Kind regards,

Abel Chemura

Academic Editor

PLOS ONE

Additional Editor Comments (if provided):

• There is no balance in the abstract in terms of the required information. More information on the results should be added and preferably also some key quantitative results. The bulk of the abstract is background and justification and yet all sections should be provided in the abstract especially the results and their implications.

• The footnotes are not the standard for referencing for PLoS ONE. Authors should strictly adhere to the referencing style of the journal by removing all footnotes and properly citing them.

• Irrigation can also be an intensification measure. A distinction between these and why in this particular context it is an adaptation strategy should be articulated.

• It is recommended that the authors stick with the standard sections of a research paper with Introduction, Methods, Results and then Discussion and Conclusion. The current presentation makes it a report and not a scientific paper. The current sections can be fused into these generic sections.

• There are no clear research questions for the analysis in the introduction which makes it difficult to see if the methods and results have answered the questions. Specific questions/objectives for each analysis may improve the readability.

• Referrals to other text should be kept at absolute minimum and as such authors should remove comments like “see next paragraph” or “see next section”. Where they want to reference tables or figures they should simply refer to the figure or table numbers and not use “see”.

• More information about the DSSAT model setting may be required such as soil profiles, genetic coefficients and management regimes that the authors used and their sources.

• There is substantial reference to the JICA internal study as a key reference in the paper. However, this may not be readily available to the readers and as such the authors should write their methods and results in an understandable manner to a person who do not have access to such internal documents. As such it is suggested that tis be removed and be rephrased for better understanding.

• Line 188-191: Adaptation should be in the absence of the measure and not in the absence of climate change. The proper evaluation framework for adaptation strategies is presented by Lobell, D. B. (2014). Global Food Security, 3(2), 72-76. Authors should consider this framework for standardisation and also for comparative analysis with similar studies.

• Appendix should be supplementary information as per PLoS ONE terminology. This should be corrected throughout the manuscript.

• The results section can be easier to understand if it follows the methods section. For example the yield forecasting, economic analysis and Runoff Analysis and Water Balance Analysis do not appear in the results sections.

• The quality of text and caption for Figure 2 and Figure 5 is very poor. Authors should improve on these.

• There is a general lack of synthesis in the discussion session which is normally presented when the Results and Discussion sections are separate. Key questions of how the findings on climate change, crop impact, hydrological change and economic aspects concur or differ with those reported elsewhere is required. It is therefore strongly suggested that the authors have a separate discussion section where they synthesise their findings in comparison with others studies and their implications for planning in East Africa or elsewhere. Reflections about replicability of this approach may also strengthen the discussion.

---

## [Author Response · Author response to Decision Letter 1]

17 Nov 2020

Thank you very much for your careful review and comments. We revised our manuscript by following your suggestions. For our specific responses, please see our cover letter.

---

## [Editor Report · Decision Letter 2]

24 Nov 2020

PONE-D-20-19964R2

Integrating Economic Measures of Adaptation Effectiveness into Climate Change Interventions: A Case Study of Irrigation Development in Mwea, Kenya

PLOS ONE

Dear Dr. Narita,

Thank you for submitting your manuscript to PLOS ONE. After careful consideration, we feel that it has merit but does not fully meet PLOS ONE’s publication criteria as it currently stands. Therefore, we invite you to submit a revised version of the manuscript that addresses the points raised during the review process.

The manuscript has significantly improved and requires a few changes before it can be accepted.

We look forward to receiving your revised manuscript.

Kind regards,

Abel Chemura

Academic Editor

PLOS ONE

Journal Requirements:

Additional Editor Comments (if provided):

-Authors should break the paragraph in Section 4.2 to at least 2 or 3 to enhance readability. Other paragraphs in the introduction and other sections are also too long and should be cut to 2 or more paragraphs.

-Authors have to remove the section on page 4 from "Our analysis reveals that climate change reduces ....to The simulation results are shown in Section 4. while Section 6 discusses them and concludes the paper. The first paragraph is not required to pre-empty the findings as this is reserved for the Results section and outlining the sections is not at all necessary.

-Authors should stick to the PLoS referencing system or at least other published papers to understand the proper referenceing for the jounral. For example, where consecutive papers are cited authors are using for e.g (10), (11), (12), (5), (13), (14) instead of the standard (5, 11-14) or even (14-15). This should be corrected throughout the paper. There is also overuse of the word e.g when referencing papers which is not appropriate way of citing papers and should be removed or used to the minimum when necessary.

-The study area and the Mwea Irrigation Development Project should come as a sub-section under Methods section. It is also not standard to have so many lines as one paragraph, break to various paragraphs. 

-There is no conclusion and authors should separate the Discussion and Conclusion provide a proper conclusion highlighting key messages and learning points from the paper. Few recommendations for building resislience will also be good for the conclusion.

---

## [Author Response · Author response to Decision Letter 2]

25 Nov 2020

We presented our responses to comments in the cover letter.

---

## [Editor Report · Decision Letter 3]

26 Nov 2020

Integrating Economic Measures of Adaptation Effectiveness into Climate Change Interventions: A Case Study of Irrigation Development in Mwea, Kenya

PONE-D-20-19964R3

Dear Dr. Narita,

We’re pleased to inform you that your manuscript has been judged scientifically suitable for publication and will be formally accepted for publication once it meets all outstanding technical requirements.

Kind regards,

Abel Chemura

Academic Editor

PLOS ONE
---

## [Editor Report · Acceptance letter]

2 Dec 2020

PONE-D-20-19964R3 

Integrating Economic Measures of Adaptation Effectiveness into Climate Change Interventions: A Case Study of Irrigation Development in Mwea, Kenya 

Dear Dr. Narita:

I'm pleased to inform you that your manuscript has been deemed suitable for publication in PLOS ONE. Congratulations! Your manuscript is now with our production department. 

Kind regards, 

on behalf of

Dr. Abel Chemura 

Academic Editor

PLOS ONE